# Chronic Alcohol Exposure Promotes Cancer Stemness and Glycolysis in Oral/Oropharyngeal Squamous Cell Carcinoma Cell Lines by Activating NFAT Signaling

**DOI:** 10.3390/ijms23179779

**Published:** 2022-08-29

**Authors:** Anthony Nguyen, Anna H. Kim, Mo K. Kang, No-Hee Park, Reuben H. Kim, Yong Kim, Ki-Hyuk Shin

**Affiliations:** 1The Shapiro Family Laboratory of Viral Oncology and Aging Research, UCLA School of Dentistry, Los Angeles, CA 90095, USA; 2UCLA Jonsson Comprehensive Cancer Center, Los Angeles, CA 90095, USA; 3Department of Medicine, David Geffen School of Medicine at UCLA, Los Angeles, CA 90095, USA; 4Laboratory of Stem Cell and Cancer Epigenetics, UCLA School of Dentistry, Los Angeles, CA 90095, USA; 5UCLA Broad Stem Cell Research Center, Los Angeles, CA 90095, USA

**Keywords:** alcohol, OSCC, cancer stem cells, glycolysis, NFAT

## Abstract

Alcohol consumption is associated with an increased risk of several cancers, including oral/oropharyngeal squamous cell carcinoma (OSCC). Alcohol also enhances the progression and aggressiveness of existing cancers; however, its underlying molecular mechanism remains elusive. Especially, the local carcinogenic effects of alcohol on OSCC in closest contact with ingestion of alcohol are poorly understood. We demonstrated that chronic ethanol exposure to OSCC increased cancer stem cell (CSC) populations and their stemness features, including self-renewal capacity, expression of stem cell markers, ALDH activity, and migration ability. The ethanol exposure also led to a significant increase in aerobic glycolysis. Moreover, increased aerobic glycolytic activity was required to support the stemness phenotype of ethanol-exposed OSCC, suggesting a molecular coupling between cancer stemness and metabolic reprogramming. We further demonstrated that chronic ethanol exposure activated NFAT (nuclear factor of activated T cells) signaling in OSCC. Functional studies revealed that pharmacological and genetic inhibition of NFAT suppressed CSC phenotype and aerobic glycolysis in ethanol-exposed OSCC. Collectively, chronic ethanol exposure promotes cancer stemness and aerobic glycolysis via activation of NFAT signaling. Our study provides a novel insight into the roles of cancer stemness and metabolic reprogramming in the molecular mechanism of alcohol-mediated carcinogenesis.

## 1. Introduction

OSCC, a common malignant tumor of the head and neck, is the sixth most common cancer worldwide with more than 50,000 new cases and 10,000 deaths in the United States alone [1,2]. Recently, incidence of OSCC among young adults has been alarmingly elevated [3,4], indicating that OSCC is an emerging public health concern.

Heavy alcohol consumption is one of the high-risk factors for cancer, and 2 to 4 percent of all cancer cases relate to alcohol consumption [5,6]. Alcohol use is associated with an increased risk of cancer, as well as with the aggressiveness of existing cancers, especially those in the head and neck, liver, and breast. Though alcohol itself may not be carcinogenic, many of its cellular effects produce likely conditions in which cells can become proliferative and invasive. Since alcohol is so widely consumed regardless of culture or social class, a detailed exploration of its molecular mechanism is necessary to counter its widespread public health effects. There are several proposed molecular mechanisms for alcohol-related carcinogenesis [7]. Acetaldehyde (AA), the major mutagenic metabolite of alcohol produced in the liver, can readily react with DNA and cause chromosomal damage to promote lesion-like growths [8,9]. Another proposed mechanism of alcohol-related carcinogenesis is through the induction of the cytochrome P450 2E1 involved in the metabolism of alcohol into AA [10] and in the breakdown of mutagens such as nitrosamines, aflatoxin, and hydrazine into their more carcinogenic counterparts [11]. However, the molecular mechanisms governing alcohol-promoted carcinogenesis remain elusive, especially the local carcinogenic effects of alcohol on tissues in closest contact on ingestion of alcohol, such as the epithelial cell lining in the oral mucosa.

Recent studies have reported the promoting effects of alcohol on tumor progression in chemically induced murine OSCC models [12,13]. Interestingly, the chemical induction of OSCC resulted in the clonal expansion of surviving stem cell populations in the basal layer of the epithelia [13], which suggests the important role of stem cell expansion in the development of OSCC. Indeed, many studies have shown the presence of CSCs (also known as tumor-initiating cells) in various human cancers, including OSCC [14]. CSCs are a subpopulation of cells within the tumor that retain certain stemness properties such as a self-renewal capacity, high mobility, and drug resistance [14]. Therefore, CSCs are responsible for tumorigenicity, metastasis, therapy resistance, and recurrence of cancers, indicating their pivotal role in tumor progression [15,16,17,18]. Accumulating evidence suggests that the expansion of CSCs and their stemness property is an important underlying mechanism for tumor progression. However, the mechanisms leading to CSC promotion are not fully understood, which has hindered the development of effective targeted therapies and the chemoprevention of cancers including OSCC.

Aerobic glycolysis (also known as the Warburg effect) has been widely recognized as a central hallmark of cancer [19]. Increased aerobic glycolysis is required for the generation of enough ATP and intermediates for macromolecular biosynthesis to meet the needs of enhanced cancer cell proliferation. Recent emerging evidence indicates that metabolic reprogramming has an important role in the maintenance of CSC populations and their stemness feature [20]. For instance, CSCs exhibit enhanced aerobic glycolytic activity compared to non-CSCs [21,22,23,24,25]. Enhanced aerobic glycolysis promoted cancer stemness properties in human cancer [26,27,28]. Conversely, inhibition of aerobic glycolysis suppressed CSC population and property in breast cancer cells [29]. However, other studies also demonstrated that slow-cycling CSCs have less aerobic glycolytic activity and higher mitochondrial oxidative phosphorylation (OXPHOS) than their differentiated progeny [30,31], indicating differential metabolic reprogramming in CSCs of different tumor types.

In addition to the link between alcohol consumption and cancer risk, emerging evidence also suggests that continued alcohol use after a cancer diagnosis may be associated with worse oncologic outcomes [32,33,34,35,36].

In this study, we explored the effects of chronic alcohol exposure on cancer stemness and aerobic glycolysis in OSCC. We also investigated the mechanism by which EtOH regulates cancer stemness and aerobic glycolysis in OSCC.

## 2. Results

### 2.1. Chronic EtOH Exposure Increases OSCC Growth In Vitro and In Vivo

To investigate the effect of alcohol on OSCC, we first examined the short-term effect of EtOH exposure (25–300 mM for 3 and 7 days) on the cell growth of human OSCC cell lines (SCC9 and UM6). Our results showed that there were no significant growth inhibitory effects at 25–150 mM concentrations (Appendix A). Since chronic alcohol consumption confers a significant risk of cancer development and progression, we sought to investigate the effect of chronic EtOH exposure on OSCC. We exposed SCC9 and UM6 to a non-cytotoxic dose of EtOH (100 mM) for extended periods and periodically examined the effect of EtOH on cell proliferation. Fifteen weeks (3 months) after the exposure, SCC9 and UM6 cells showed enhanced cell proliferation; thus, they were denoted SCC9/EtOH and UM6/EtOH, respectively (Figure 1A). Importantly, it should be noted that after 15 weeks of exposure, we withdrew EtOH from the culture medium and performed all the experiments in the absence of EtOH. By doing so, we wanted to exclude the immediate effect of EtOH on the biological behaviors of tested OSCC cells. We further examined the effect of chronic EtOH exposure on malignant growth properties, such as anchorage-independence growth. A soft agar assay revealed that the anchorage-independent growth ability of SCC9/EtOH and UM6/EtOH was much greater than their corresponding untreated controls, SCC9 and UM6, respectively (Figure 1B). Since the ability of anchorage-independence growth has been linked to tumor cell aggressiveness in vivo, including tumorigenicity [34], we performed a xenograft tumor assay to measure tumorigenicity. As shown in Figure 1C, SCC9 and UM6 cell lines failed to form tumors in nude mice, indicating that they are not full tumorigenic or weakly tumorigenic [37,38]. They also displayed different tumor-forming/growth kinetics in the animal. SCC9/EtOH and UM6/EtOH developed tumors faster than their controls, SCC9 and UM6, respectively, and the sizes of tumors were greater than those developed from their controls. In addition, in vivo tumor growth of SCC9/EtOH and UM6/EtOH sustained longer periods than those of the controls. Nevertheless, tumors formed by the EtOH-exposed cells completely regressed 2 months after the inoculation (data not shown). Taken together, our data suggest that chronic exposure to EtOH increases OSCC growth in vivo, an indication of increased tumorigenic potential. However, the chronic EtOH exposure failed to convert the tested weakly tumorigenic OSCC cells to fully tumorigenic in the xenograft assay.

### 2.2. Chronic EtOH Exposure Increases the Number of ALDH1^HIGH^ Stem-like Cell Populations and Characteristics Associated with Stemness in OSCC

Since emerging evidence indicates that expansion of CSCs and their stemness property are critically implicated in OSCC progression [14], we further explored the effect of EtOH on the CSC phenotype in OSCC. The activity of aldehyde dehydrogenase 1 (ALDH1) has been widely used as a marker for isolating CSCs. Moreover, ALDH1^HIGH^ cancer cells displayed higher CSC properties compared to ALDH1low cells, indicating that the ALDH1^HIGH^ cells are CSC-enriched populations [39,40,41]. We sorted ALDH1^HIGH^ and ALDH1low cells from the EtOH-exposed OSCC and their control cells by performing flow cytometry analysis. The assay revealed a significant increase in the ALDH1^HIGH^ cell population in SCC9/EtOH and UM6/EtOH compared to their controls (Figure 2A,B). Moreover, gene expression of ALDH1 and other CSC markers, including CD44 [42] and CD133 [43], were significantly upregulated in the EtOH-exposed OSCC cells (Figure 2C), indicating that chronic EtOH exposure to OSCC increases CSC populations. Many studies have demonstrated that self-renewal capacity and increased migration ability are important characteristics of CSCs [14]. Compared to the untreated controls, SCC9/EtOH and UM6/EtOH showed a robust increase in self-renewal (Figure 2D) and migration capacity (Figure 2E). CSC factors related with self-renewal and migration capacity were also increased in the EtOH-exposed cell compared to their controls (Figure 2F). However, such changes were not detected by acute EtOH exposure (data not shown). Taken together, our data indicate that chronic exposure to alcohol expands CSC populations and promotes their stemness property in OSCC.

### 2.3. Chronic EtOH Exposure Promotes Aerobic Glycolysis in OSCC

Aerobic glycolysis (also known as the Warburg effect) is a unique metabolic phenotype of cancer cells that requires enough ATP and intermediates for macromolecular biosynthesis to meet the needs of enhanced cell proliferation [19]. Moreover, recent emerging evidence indicates that increased aerobic glycolytic activity is required to maintain CSCs in human cancer [20]. Thus, to assess the effect of chronic EtOH exposure on aerobic glycolysis in OSCC, we compared key aerobic glycolytic activities (i.e., glucose uptake and lactate secretion) of the control and EtOH-exposed OSCC cells. Both glucose uptake (Figure 3A) and lactate secretion (Figure 3B) were significantly increased in SCC9/EtOH and UM6/EtOH compared to their untreated controls. We also tested the acute effect of EtOH on glycolysis and found that acute EtOH exposure had no significant effects on glucose uptake and lactate secretion in OSCC (Appendix A). To further confirm the increased glycolysis in the EtOH-exposed cells, we measured gene expression of glycolytic enzymes and found that multiple key glycolytic enzymes (i.e., GP1, TPI1, ENO1, and PKM2) were robustly increased in SCC9/EtOH and UM6/EtOH (Figure 3C) compared to their controls. Collectively, our findings indicate that chronic EtOH exposure to OSCC promotes aerobic glycolysis, suggesting a functional link between EtOH-promoted cancer stemness and aerobic glycolysis.

### 2.4. Increased Aerobic Glycolysis Is Required to Maintain the Stemness Characteristics of EtOH-Exposed OSCC Cells

Next, to investigate whether chronic EtOH exposure promotes the CSC phenotype of OSCC by increasing aerobic glycolysis, we utilized 2-deoxy-glucose (2-DG), a glycolysis inhibitor, to suppress the glycolytic activity of OSCC cells. We found that there was minimal cytotoxic effect of 2-DG at 1–5 mM concentrations for 2 and 4 days (Appendix A). Using these concentrations, we demonstrated that 2-DG suppressed the aerobic glycolytic activity of the EtOH-exposed OSCC cells (Appendix A). Furthermore, 2-DG inhibited self-renewal (Figure 4A) and the migration ability (Figure 4B) of the EtOH-exposed cells in a dose-dependent manner. Interestingly, the magnitude of cancer stemness suppression in the EtOH-exposed cells was significantly greater than that in their untreated controls. These indicate that the CSC phenotype of the EtOH-exposed cells is more dependent on glycolysis than those of the control cells, suggesting that increased glycolysis is required to support the CSC phenotype in the EtOH-exposed OSCC cells. Taken together, our findings conclude that chronic EtOH exposure promotes the cancer stemness of OSCC by increasing aerobic glycolysis.

### 2.5. Chronic EtOH Exposure Activates NFAT Signaling

Emerging evidence indicates a dual role of NFAT signaling in cancer stemness and glycolysis [44,45], further suggesting that NFAT signaling could be a potential molecular player responsible for the EtOH-induced events. Thus, to gain an insight into the mechanism by which EtOH promotes cancer stemness and aerobic glycolysis in OSCC, we explored the role of NFAT signaling in the EtOH-induced phenotypic changes. First, we investigated the effect of chronic EtOH exposure on NFAT activity using a luciferase reporter vector under the control of a chimeric promoter containing three adjacent canonical NFAT binding sites [46]. Thus, luciferase activity is indicative of NFAT activity. The assay revealed that the luciferase activity is significantly increased in SCC9/EtOH and UM6/EtOH compared to their controls (Figure 5A). Given that chronic EtOH exposure activates NFAT activity, we also measured the expression of NFAT downstream target genes in the EtOH-exposed OSCC cells. Well-known downstream targets of NFAT, such as IL2, IL4, IL6, and TNFα, were significantly upregulated in the EtOH-exposed cells (Figure 5B), indicating the activation of NFAT signaling by chronic EtOH exposure.

Since there are multiple NFAT members (NFATc1-c4) identified [47], we further investigated which of the NFAT members are involved in the EtOH-induced NFAT activation. We found a significant upregulation of NFATc2 in both SCC9/EtOH and UM6/EtOH compared to their corresponding controls (Figure 5C). Moreover, NFATc2 was primarily found in the cytoplasm of untreated controls but accumulated both the cytoplasm and the nucleus of EtOH-exposed OSCC cells (Figure 5D), indicative of NFATc2 activation by chronic EtOH exposure. NFATc1 and NFATc3 were also increased in SCC9/EtOH but not in UM6/EtOH compared to their untreated controls (Figure 5C). NFATc4 was not detected in the OSCC cells. Collectively, our results indicate the activation of NFATc2 by chronic EtOH exposure in OSCC.

### 2.6. Silencing NFATc2 Inhibits Cancer Stemness and Aerobic Glycolysis in EtOH-Treated OSCC

Next, to investigate whether EtOH-induced NFATc2 activation contributes to the increase in CSC phenotype and aerobic glycolysis in OSCC, we suppressed NFATc2 using siRNA in the EtOH-exposed OSCC cells (Figure 6A and Appendix A). Suppression of NFATc2 also resulted in decreased NFAT downstream targets (Figure 6B), indicating functional suppression of NFATc2 by siRNA. Inhibition of NFATc2 led to a significant decrease in self-renewal (Figure 6C) and migration ability (Figure 6D) both in SCC9/EtOH and UM6/EtOH, indicating that increased NFATc2 is required to maintain the EtOH-increased stemness phenotype in OSCC. We also examined the effect of NFATc2 inhibition on glycolysis and found that NFATc2 inhibition resulted in a significant suppression in glucose uptake (Figure 7A) and lactate secretion (Figure 7B) in SCC9/EtOH and UM6/EtOH, indicating that increased NFATc2 is required to maintain EtOH-induced glycolytic activity in OSCC. Taken together, our results indicate that activation of NFATc2 is required to maintain increased cancer stemness and aerobic glycolysis in the EtOH-exposed OSCC cells. Further studies using the NFAT antagonist, cyclosporine A (CsA) revealed that chemical inhibition of NFAT also inhibited both CSC properties and glycolysis in the EtOH-exposed OSCC cells (Appendix A). Our findings also indicate a novel dual role of NFAT in the regulation of glucose metabolism and cancer stemness in OSCC.

## 3. Discussion

In this study, we demonstrated that chronic EtOH exposure enhanced tumor growth and the malignant property of OSCC, indicating that chronic EtOH exposure could play a role in promoting the progression of OSCC. Interestingly, we also found that chronic EtOH exposure increased aerobic glycolysis and the stemness phenotype of OSCC, and the increase in glycolysis was required to maintain the EtOH-induced stemness phenotype of OSCC cells. Subsequently, we found that NFAT signaling was activated by chronic EtOH exposure. Chemical inhibition of NFAT signaling suppressed EtOH-induced glycolysis and cancer stemness. Last, we demonstrated that silencing NFATc2 successfully repressed these two EtOH-induced events in OSCC, suggesting a dual role of NFATc2 in the regulation of glycolysis and cancer stemness. Hence, our results offer a novel insight into the potential mechanism of alcohol-mediated cancer progression.

EtOH concentration of 20 mM represents a blood alcohol concentration of 0.08%. Although the use of 100 mM EtOH could be deemed high, a high concentration of EtOH was used to investigate the direct effects of EtOH, which could represent a local level of alcohol in the oral cavity as opposed to a diluted level of alcohol in the bloodstream. Obviously, alcohol consumption is not as long as a duration of 3 months. However, to ensure the effects of chronic alcohol were not fleeting, we selected cells that were tolerant to the EtOH concentration (100 mM) and the duration of the EtOH exposure. The dose of 100 mM EtOH did not induce a significant reduction in OSCC growth for both short-term and long-term exposure.

The Warburg effect, also known as aerobic glycolysis, is a classical hallmark of cancer metabolism and enables cancer cells to satisfy the energy demanded for rapid cell growth and division [19]. Indeed, many studies demonstrated increased aerobic glycolysis in multiple cancer types, including OSCC [48]. Interestingly, the involvement of alcohol in the process of glycolysis has been shown. For instance, alcohol administration increased the activity of glycolytic enzymes and the contents of glucose and lactate in the rat brain in vivo [49]. A similar observation was also found in mouse auditory cells in vitro, clearly indicating the promoting effects of alcohol on glycolysis [50]. However, the effect of alcohol on glycolysis in human cells, especially OSCC cells, has yet to be documented. Our study revealed that chronic EtOH exposure in OSCC cells promoted aerobic glycolysis as demonstrated by increased glucose uptake, lactate production, and glycolysis-related gene expression. As demonstrated by 2-DG, the glycolysis inhibitor, the EtOH-exposed OSCC cells were more dependent on glycolysis than their control cells for their growth, suggesting that the increased glycolysis is required for the enhanced proliferation capacity caused by chronic ethanol exposure. These findings agree with the Warburg effect.

Animal studies have demonstrated the promoting effect of alcohol on tumor progression in 4-nitroquinoline-1-oxide (4NQO)-induced murine tongue cancer models [12,13]. Interestingly, the chemical induction of OSCC resulted in the expansion of surviving stem/progenitor cell population, which suggests the importance of stem cell expansion for oral cancer development [13]. Moreover, compared to 4NQO administration alone, alcohol administration combined with 4NQO increased the active form of β-catenin [13], which is a key regulator for the cancer stemness pathway [51]. These data suggest that alcohol may promote OSCC progression by increasing CSCs. Indeed, our study demonstrated that chronic EtOH exposure to OSCC increased ALDH1^HIGH^ CSC population. The increased CSC population was accompanied with promoted CSC properties, including, i.e., self-renewal, migration, anchorage-independent growth, and tumorigenicity. Our findings are consistent with previous studies demonstrating the promoting effects of EtOH on CSC population and phenotype in multiple cancer types, such as breast [52,53] and liver [54,55].

Studies have indicated essential roles of metabolic reprogramming for the genesis and maintenance of CSC population and phenotype. CSCs exhibit a higher aerobic glycolytic activity compared to their corresponding non-CSCs [21,22,23,24,25,56]. Inhibition of glycolysis decreased CSC population and property [29,57]. CSCs also expressed elevated levels of glycolytic enzymes such as Glut-1, HK, G6PD, PDK1, PKM2, and LDH [21,23,56]. Moreover, ectopic expression of glycolytic enzymes enhanced CSC phenotype [58,59]. These data indicate that increased aerobic glycolysis is required to support the CSC phenotype. In our study, we showed that EtOH increased not only aerobic glycolysis, but also the CSC phenotype in OSCC. In agreement with previous reports, 2-DG inhibited glycolysis and key CSC properties such as self-renewal and the migration capacity of the EtOH-exposed OSCC cells. Moreover, the CSC phenotype of the EtOH-exposed OSCC cells is more dependent on glycolysis than those of the control cells, indicating that the EtOH-induced glycolysis is required to maintain the CSC phenotype in the EtOH-exposed cells. Thus, we hypothesize that chronic EtOH exposure promotes cancer stemness of OSCC by increasing aerobic glycolysis. However, molecular mechanisms underlying the EtOH-induced events are not well-understood. To further broaden our understanding of the role of alcohol on oral cancer progression, the effect of chronic EtOH exposure to immortalized and non-tumorigenic human oral keratinocytes, i.e., OKF6/tert and HOK-16B, should warrant investigation [60].

NFAT signaling plays an oncogenic role in multiple cancer types by promoting cancer stemness. For instance, NFATc2 promoted cancer stemness of colorectal cancer via AJUBA-mediated YAP activation [61]. NFATc2 enhanced the CSC phenotype through the NFATc2/Sox2ALDH axis in lung adenocarcinoma [62]. We reported that NFATc3 plays an oncogenic role in OSCC by promoting cancer stemness via expression of Oct4. Interestingly, recent studies have also demonstrated the significant role of NFAT in metabolic reprogramming in various human cancers. NFATc1 is overexpressed in prostate cancer cells, and its inhibition suppresses aerobic glycolysis, concurrent with a decrease of pyruvate kinase 2 (PKM2) [45]. NFAT5 is upregulated in pancreatic cancer cells and promotes aerobic glycolysis by inducing phosphoglycerate kinase 1 (PGK-1) [44]. Moreover, HIF1α, the key transcription factor involved in glycolysis, contains a consensus NFAT binding site in its promotor region, and its transcription is activated by NFAT [63]. Collectively, these clearly indicate the potential dual roles of NFAT signaling in glycolysis and cancer stemness. However, the role of NFAT in glycolytic activity in OSCC remains largely unknown.

Our study demonstrated that NFAT signaling is activated by chronic EtOH exposure in OSCC. The activation was supported by observation showing augmented NFAT activity with the induction of a subset of NFAT downstream targets. Moreover, among NFAT members, NFATc2 was consistently upregulated by chronic EtOH exposure in multiple OSCC cell lines. Silencing NFATc2 in the EtOH-exposed OSCC cells significantly diminished not only the stemness phenotype but also glycolytic activity, indicating the dual role of NFAT activation in the two EtOH-induced events in OSCC. Chemical inhibition of NFAT also suppresses both EtOH-increased glycolysis and cancer stemness in OSCC. Thus, we conclude that activation of NFAT signaling by chronic EtOH exposure results in an increase in the aerobic glycolysis and stemness phenotype of OSCC. Therefore, NFAT could be an effective therapeutic target for alcohol-related cancer. However, the underlying molecular mechanism of how NFAT signaling regulates these two processes warrants investigation.

In our study, we found that silencing NFATc2 in the EtOH-exposed OSCC cells resulted in a decrease in multiple genes involved in glycolysis and cancer stemness, including HIF1α, TP1, ENO1, PKM2, ALDH1A, Bmi1, and Oct4 (data not shown). Furthermore, the expression of these genes was increased by chronic EtOH exposure in OSCC, suggesting that they could be potential candidates to investigate downstream targets of NFATc2-regulated glycolysis and cancer stemness. Indeed, studies revealed that HIF1α and Oct4 are the transcription targets of NFAT [62,63,64]. We also found the presence of multiple NFATc2 consensus binding sites (5′-GGAAA-3′) in the promoter region of PKM2 (data not shown). Therefore, further investigation is necessary to identify the downstream targets of the EtOH-NFATc2 axis that are responsible for EtOH-induced events, such as glycolysis and cancer stemness, in OSCC. Overall, our study demonstrates that chronic alcohol exposure promotes tumor progression by enhancing glycolysis and cancer stemness via activation of NFAT signaling in OSCC.

## 4. Materials and Methods

### 4.1. Cell Culture and Reagent

Two human OSCC cell lines, SCC9 and UM6, were cultured in DMEM/F12 (LifeTechnologies, Carlsbad, CA, USA) supplemented with 10% serum (Gemini Bioproducts, West Sacramento, CA, USA), 0.4 μg/mL hydrocortisone (Sigma-Aldrich, St. Louis, MO, USA), and 5 μg/mL Gentamycin aminoglycoside antibiotic (Invitrogen, Waltham, MA, USA). All cell lines were grown in a humidified incubator with 5% CO_2_ at 37 °C. Cells were cultured in the culture medium containing the indicated ethanol concentration, and the medium was changed every 3–4 days. Absolute ethanol was purchased from Fisher Bioreagents. 2-Deoxy-D-glucose (2-DG) was purchased from Sigma-Aldrich. All cell lines were routinely tested and authenticated using cell morphology, proliferation rate, a panel of genetic markers, and contamination checks. All cell lines were also tested for mycoplasma, using the MycoAlert detection kit (Cambrex, East Rutherford, NJ, USA) and shown to be negative.

### 4.2. Cell Proliferation Assay

Cell growth was determined by using the tetrazolium salt (MTT) cell proliferation assay kit (ATCC, Manassas, VA, USA) and cell counting. The cells were plated at 2 × 10^3^ cells per well into a 96-well plate. They were then incubated in a culture medium containing various ethanol concentrations (0–300 mM) for 2 and 7 days. Absorbance at 570 nm was determined using a microplate reader. For cell counting, the cells were plated at 2 × 10^4^ cells per well into a 6-well plate. They were then incubated for indicated days and counted.

### 4.3. Anchorage-Independent Growth

To determine colony-forming efficiency in a semi-solid medium, 1 × 10^4^ cells were plated in culture medium containing 0.3% agarose over a base layer of serum-free medium containing 0.5% agarose. Three weeks after incubation, colonies were counted. The experiment was performed in the absence of ethanol and in triplicates with 60 mm dishes.

### 4.4. In Vivo Xenograft Tumor Assay

Five million cells were subcutaneously injected into the flank of immunocompromised mice (strain *nu*/*nu*, Charles River Laboratories, Wilmington, MA, USA). Five immunocompromised mice (female, 6 to 8 weeks old) per group were used and there were four groups: mice injected with SCC9, SCC9/EtOH, UM6, and UM6/EtOH. The animal study was performed according to the protocol approved by the UCLA Animal Research Committee. The kinetics of the tumor growth were determined by measuring the volume in three perpendicular axes of the nodules using micro-scaled calipers.

### 4.5. Glucose Uptake and Lactate Secretion

Glucose uptake was measured with a Glucose-Glo™ assay kit (Promega, Madison, WI, USA). Lactate secretion was measured with Lactate-Glo™ assay (Promega). Measurements were normalized to cell number.

### 4.6. Quantitative Real-Time PCR (qPCR)

cDNA was synthesized from 5 μg of total RNA using the SuperScript first-strand synthesis system (Invitrogen). Then, qPCR was performed using a PowerUp SYBR Green Master Mix (Thermo Fisher Scientific, Waltham, MA, USA) and QuantStudio 3 qPCR System (Thermo Fisher Scientific) as described in our prior work [64]. The primer sequences were obtained from the Universal Probe Library (Roche, Basel, Switzerland) and the sequences can be made available upon request. A second derivative Cq value determination method was used to compare fold-differences according to the manufacturer’s instructions.

### 4.7. ALDH1 Assay

Using an Aldehyde Dehydrogenase-Based Cell Detection Kit (STEMCELL, Vancouver, Canada), the ALDH enzymatic activity was determined. A total of 1 × 10^6^ cells were re-suspended in the Aldeflour assay buffer in a volume of 1 mL. Fluorescent nontoxic Aldeflour Reagent BODIPY™ (1.25 µL) was added as a substrate to measure ALDH enzymatic activity in intact cells. Immediately after adding the substrate reagent, 0.5 mL of the cell suspension was transferred into the control tube that contains the specific inhibitor for ALDH, diethylaminobenzaldehyde (DEAB) for calculating background fluorescence. Then, cells were incubated at 35 °C for 30 min and fluorescence data acquisition was made by using a BD FACScan flow cytometer (BD Biosciences, East Rutherford, NJ, USA).

### 4.8. Tumor Sphere Formation Assay

Three thousand cells were grown in 3 mL of serum-free DMEM/F12 media supplemented with 1:50 B27 (Invitrogen), 20 ng/mL EGF, 20 ng/mL, 10μg/mL insulin, penicillin, streptomycin, and amphotericin B in ultra-low attachment 6-well plates (Corning, Corning, NY, USA). SCC9 and its derivatives were incubated for 6 days, and UM6 and its derivatives were cultured for 10 days. The number of tumor spheres formed was observed and counted under a microscope. The experiment was performed in the absence of ethanol.

### 4.9. Migration Assay

Cell migration was measured using 6.5 mm transwell chambers with 8.0 μm polycarbonate membranes (Corning: Product#3422) as described in our previous publication [65]. The experiment was performed in the absence of ethanol.

### 4.10. Luciferase Reporter Assay

Transfection and a luciferase assay were carried out as described in prior work [66]. Briefly, cells were transfected with the pGL3-NFAT-luc vector for 24 h. Cells were then harvested, and luciferase activity was measured using a dual luciferase reporter assay system (Promega). For the normalization of transfection efficiency, the cells were also co-transfected with pRL-SV40 (Promega) containing the *renilla* luciferase gene under SV40 promoter.

### 4.11. Confocal Laser Scanning Microscopy

Five thousand cells were seeded on the four chamber slides (Thermo Fisher Scientific) one day prior to the immunofluorescence staining. After cell permeabilization and blocking, cells were probed with NFATc2 primary antibody overnight, then with Alexa Fluor 594 dye-conjugated secondary antibody and DAPI (blue-green) for confocal laser scanning. Confocal laser scanning microscopy was performed using a Fluoview FV10i Confocal Microscope (Olympus, Tokyo, Japan) and images were captured with 60X oil objective under different gain settings. The 559 nm laser diode was used to capture NFATc2 staining, and the 405 nm laser diode was used to capture the DAPI nuclear stain. Image acquisition and further adjustment of brightness was performed using an Olympus FluoView FV10ASW Version 4.2a software. Fluorescent images of cells were taken as single channel images and then converted to overlay images and all images were saved in TIFF format.

### 4.12. Small Interfering RNA (siRNA) Transfection

NFATc2 siRNA (sc-36055; Santa Cruz Biotechnology, Dallas, TX, USA) and control siRNA (sc-37007; Santa Cruz Biotechnology) were purchased and introduced into cells using Lipofectamine RNAiMAX (Invitrogen). Cells (2 × 10^5^) were plated in 60 mm dishes and transfected with 10 μg siRNA. The cultures were harvested after one day post-transfection for expression and functional analyses.

### 4.13. Statistical Analysis

The statistical analyses were calculated using GraphPad Prism 5. The data were expressed as mean ± standard deviation (SD). Data between two groups were compared using parametric Student’s *t*-test or paired *t*-test. A value of *p* < 0.05 was considered as statistically significant.

## 5. Conclusions

We demonstrate that chronic alcohol exposure enhances OSCC progression by increasing cancer stemness and aerobic glycolysis via activation of NFAT signaling. Thus, our study provides a novel insight into the roles of cancer stemness and metabolic plasticity in the molecular mechanism of alcohol-mediated carcinogenesis.

## Figures and Tables

**Figure 1 ijms-23-09779-f001:**
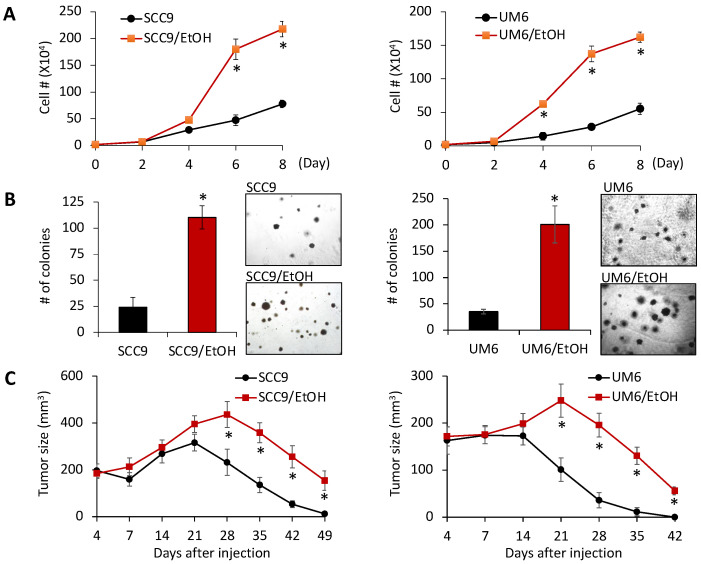
Chronic EtOH exposure increases tumor growth of OSCC both in vitro and in vivo. (**A**) Effect of chronic EtOH exposure on proliferation capacity of OSCC was determined by cell counting. Passage-matched controls, SCC9 and UM6, were used to compare with SCC9/EtOH and UM6/EtOH, respectively. The assay was performed in the absence of EtOH. * *p* < 0.001 compared to untreated controls by Student’s *t*-test. (**B**) Effect of chronic EtOH exposure on anchorage independent growth ability of OSCC was determined by soft agar assay. The assay was performed in the absence of EtOH. Data are means ± SD of triplicate experiments. * *p* < 0.001 compared to untreated controls by Student’s *t*-test. (**C**) Effect of chronic EtOH exposure on in vivo tumorigenicity of OSCC was determined by xenograft tumor assay. * *p* < 0.001.

**Figure 2 ijms-23-09779-f002:**
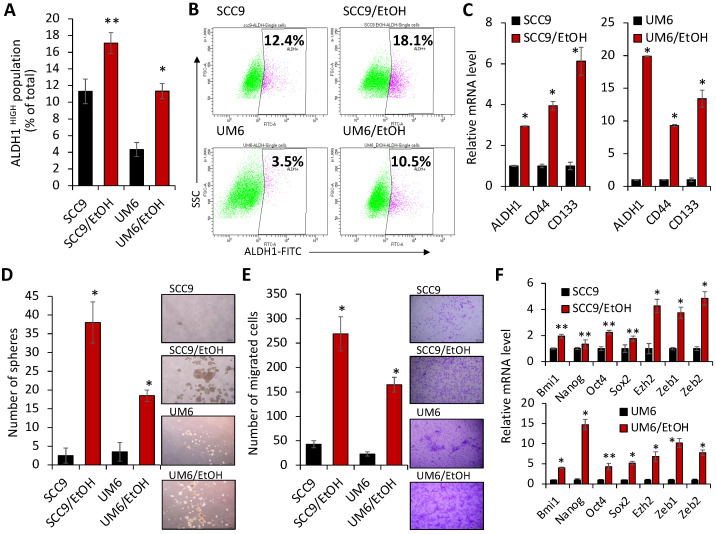
Chronic EtOH exposure enhances CSC population and property of OSCC. (**A**) Effect of chronic EtOH exposure on ALDH1^HIGH^ CSC population in OSCC was determined by Aldefluor flow cytometry-based assay. Data are means ± SD of three independent assays. * *p* < 0.001, ** *p* < 0.05. (**B**) Representative images of Aldefluor flow cytometry-based assay. The number shown in each panel reflects the percentage of ALDH1^HIGH^ cells in each cell type. (**C**) Effect of chronic EtOH exposure on CSC markers (ALDH1A, CD44, and CD133) in OSCC was determined by qPCR. * *p* < 0.001. (**D**) Effect of chronic EtOH exposure on self-renewal capacity in OSCC was measured by tumor sphere formation assay. Representative images of tumor spheres formed by the OSCC cells were shown on the right of bar graph. Data are means ± SD of triplicate experiment. * *p* < 0.001. (**E**) Effect of chronic EtOH exposure on migration ability in OSCC was determined by transwell migration assay. Migration ability was described as number of migrated cells per field with data as mean ± SD for three randomly selected fields. Representative images of transwell migration assay are shown on the right of bar graph. * *p* < 0.001 (**F**) Effect of chronic EtOH exposure on stemness-related genes in OSCC was determined by qPCR. * *p* < 0.001, ** *p* < 0.05.

**Figure 3 ijms-23-09779-f003:**
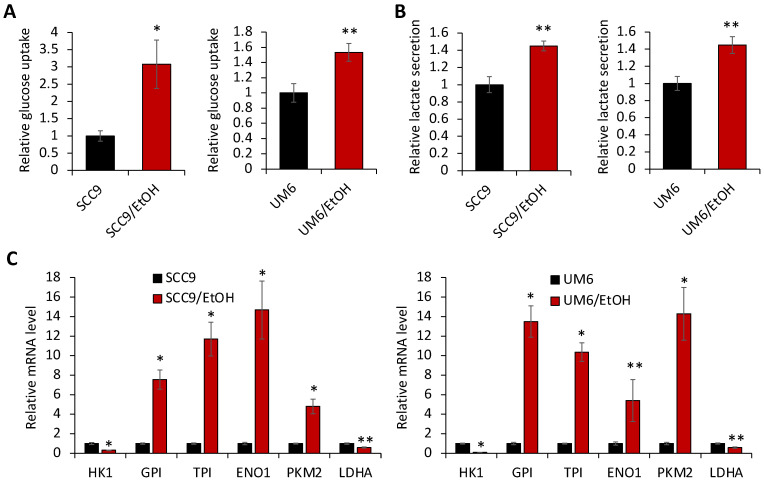
Chronic EtOH exposure increases the glycolytic activity of OSCC. (**A**) Effect of chronic EtOH exposure on glucose uptake by OSCC was determined using a Glucose-Glo™ assay kit (Promega). Relative glucose uptake by the EtOH-exposed cells (SCC9/EtOH and UM6/EtOH) were plotted as fold induction against those in their corresponding untreated controls (SCC9 and UM6). Data are means ± SD of triplicate experiments. (**B**) Effect of chronic EtOH exposure on lactate secretion by OSCC was determined using Lactate-Glo™ assay (Promega). * *p* < 0.001, ** *p* < 0.05. (**C**) Effect of chronic EtOH exposure on glycolytic gene expression in OSCC was determined by qPCR. Levels of the glycolytic genes were normalized with the level of GAPDH. Their levels in the EtOH-exposed cells (SCC9/EtOH and UM6/EtOH) were plotted as fold induction against those in their corresponding untreated controls (SCC9 and UM6). * *p* < 0.001, ** *p* < 0.05.

**Figure 4 ijms-23-09779-f004:**
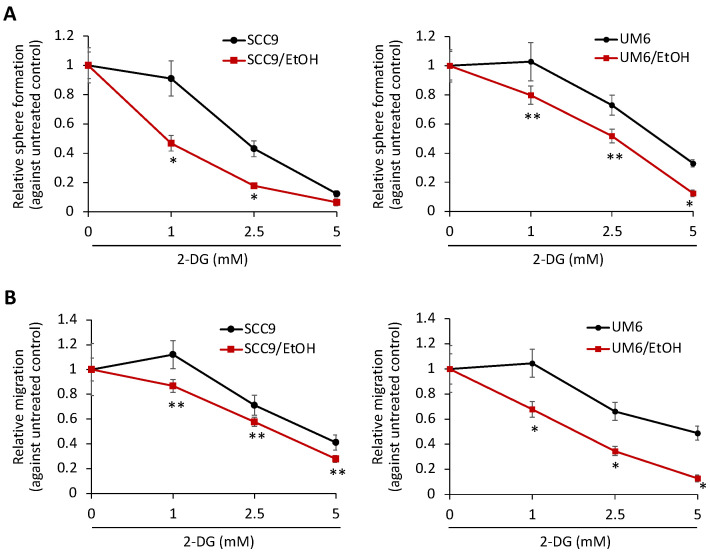
EtOH-exposed OSCC cells depend on glycolysis for their stemness property. (**A**) Effect of glycolysis inhibitor 2-DG on self-renewal capacity of EtOH-exposed OSCC and their untreated controls were determined by tumor sphere formation assay. The assay was performed in the tumor sphere medium containing indicated 2-DG concentrations (0–5 mM), and the numbers of tumor spheres were counted. * *p* < 0.001, ** *p* < 0.05. (**B**) Effect of 2-DG on migration capacity of EtOH-exposed OSCC and their untreated controls were determined by transwell migration assay.

**Figure 5 ijms-23-09779-f005:**
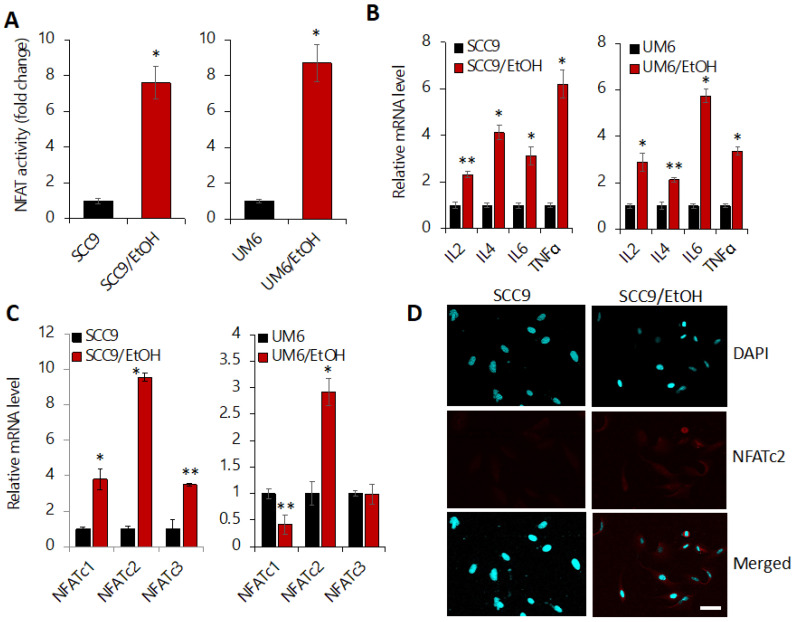
Chronic EtOH exposure activates NFAT signaling in OSCC. (**A**) Effect of chronic EtOH exposure on NFAT activity was determined by luciferase promoter assay. Cells were transfected with pGL3-NFAT-luc vector whose luciferase activity is under the control of a chimeric promoter containing three adjacent canonical NFAT binding sites. Cells were also transfected with pRL-SV40 for normalization of transfection efficiency. * *p* < 0.001. (**B**) Effect of chronic EtOH exposure on the expression of NFAT downstream targets (IL2, IL4, IL6, and TNFα) in OSCC was determined by qPCR. * *p* < 0.001, ** *p* < 0.05. (**C**) Effect of chronic EtOH exposure on the expression of NFAT isoforms (NFATc1, NFATc2, and NFATc3) in OSCC was determined by qPCR. * *p* < 0.001, ** *p* < 0.05. (**D**) Effect of chronic EtOH exposure on the intracellular localization of NFATc2 was determined by confocal laser scanning microscopy. SCC9 has NFATc2 immunofluorescence staining (red) mainly in the cytoplasm while SCC9/EtOH has stronger staining both in the cytoplasm and the nucleus, which indicates increased NFATc2 expression, as well as dominant nuclear translocation. Bar, 20 μm.

**Figure 6 ijms-23-09779-f006:**
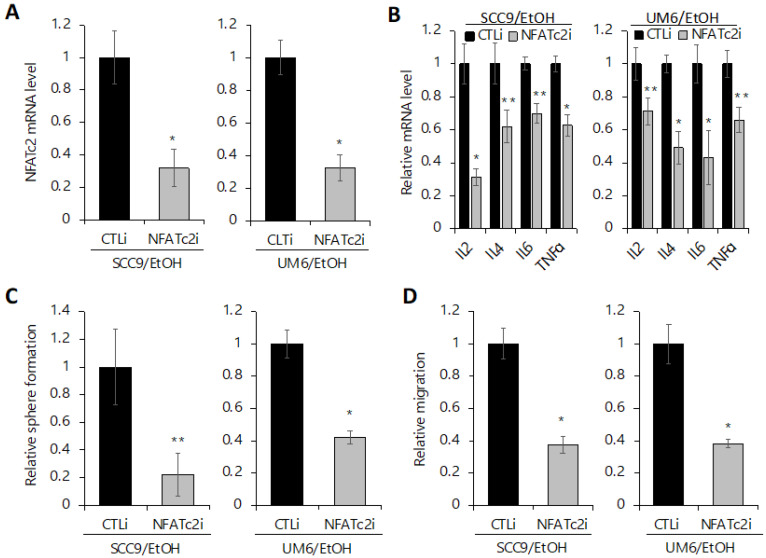
Knockdown of NFATc2 suppresses the stemness property of the EtOH-exposed OSCC. (**A**) NFATc2 in the EtOH-exposed OSCC cells (SCC9/EtOH and UM6/EtOH) was knocked down using siRNA against NFATc2 (NFATc2i). The cells transfected with control siRNA (CTLi) were included for comparison. NFATc2 knockdown was confirmed by qPCR. * *p* < 0.001 compared to CTLi-transfected controls by paired *t* test. (**B**) Effect of NFATc2 knockdown on the expression of NFAT downstream targets (IL2, IL4, IL6, and TNFα) was determined by qPCR. * *p* < 0.001, ** *p* < 0.05. (**C**) Effect of NFATc2 knockdown on self-renewal capacity of the EtOH-exposed OSCC cells was determined by tumor sphere formation assay. * *p* < 0.001, ** *p* < 0.05. (**D**) Effect of NFATc2 knockdown on migration capacity of the EtOH-exposed OSCC cells was determined by transwell migration assay, respectively. * *p* < 0.001.

**Figure 7 ijms-23-09779-f007:**
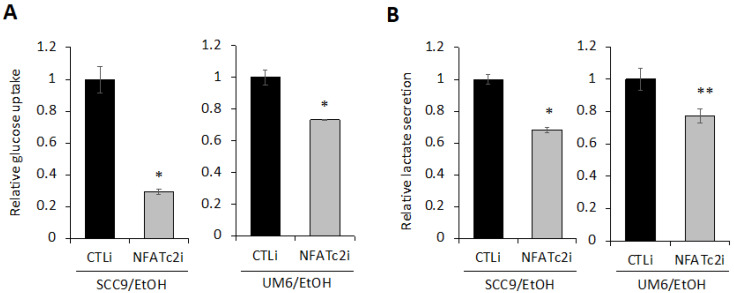
Knockdown of NFATc2 inhibits the glycolytic activity in the EtOH-exposed OSCC. (**A**) Effect of NFATc2 knockdown on glucose uptake in the EtOH-exposed OSCC cells (SCC9/EtOH and UM6/EtOH) was determined using a Glucose-Glo™ assay kit. (**B**) Effect of NFATc2 knockdown on lactate secretion in the EtOH-exposed OSCC cells was determined using Lactate-Glo™ assay. * *p* < 0.001 and ** *p* < 0.05.

## Data Availability

All data are available in the text and Appendix A.

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
