# Peer review of "Chronic Alcohol Exposure Promotes Cancer Stemness and Glycolysis in Oral/Oropharyngeal Squamous Cell Carcinoma Cell Lines by Activating NFAT Signaling"

_ijms, 2022, doi:10.3390/ijms23179779_

Round 1

Reviewer 1 Report

This manuscript that submitted by Shin et al. aimed to reveal the mechanisms attributing to chronic alcohol consumption on oral cancer progression. The authors examined the long-term effect of alcohol treatment on oral cancer stem-like properties and aerobic glycolysis. The authors found that long-term treatment of alcohol increased NFATc2 activity and promote the downstream gene expression, while knockdown NFATc2 repressed gene expression. However, this manuscript has many flaws needing to be solve. There are some suggestions for the authors.

Major concerns

1. The aim of present study focused on chronic alcohol consumption on oral cancer progression. But, alcohol is a well-known potential carcinogen, what the rationale for study the cancer promoting effect of alcohol on oral cancer rather than prevention? I'm curious how likely it is that a patients with oral cancer will continue to drink alcohol. The authors should provide more reasonable motivation in the manuscript.

2. In the introduction, the ref. 3 is not relevant citation. the authors should revise it.

3. The overall manuscript, the stemness transcriptional factors and CD markers, glycolysis signaling and cytokines are just examined at gene level, I am not sure weather the genes expression level are consistent with protein level. In addition, the siRNA knockdown efficacy should further be examined by western blotting.

4. The UM6 is not human oral cancer cell line. In present study, the authors just apply one human oral cancer cell line. Using one cell line is hard to claim the phenomenon which influences by alcohol.

5. In the fig.1, the colony formation pictures seemed to not consistent with its quantitative results. Moreover, why the SCC9 and UM6 tumors growth shrink after one month. Do these two cancer cell lines hard to form solid tumors in mouse model?

6. The authors want to demonstrate that alcohol promotes cancer stem-like properties and aerobic glycolysis via activation of NFATc2, but the authors just examine NFATc2 downstream genes being reduced by siRNA, lacking the rescue  experiments for opposite demonstration. In addition, I am curious weather NFATc2 expression level is up-regulated in oral cancer patients with chronic alcohol consumption history comparing to non-alcohol drink patients? Have the authors ever survey the clinical relevant of NFATc2 expression level on oral cancer patients with alcohol consumption?

Reviewer 2 Report

Nguyen et al demonstrate an interesting link between chronic ethanol exposure to OSCC and the increased cancer stem cell (CSC) populations and their stemness features. The authors study their self-renewal capacity, expression of stem cell markers, ALDH activity, and migration ability. The results show that ethanol exposure also led to a significant increase in aerobic glycolysis. Moreover, increased aerobic glycolytic activity was required to support the stemness phenotype of ethanol-exposed OSCC, suggesting a molecular coupling between cancer stemness and metabolic reprogramming. Alcohol consumption is always accepted as a high-risk health factor though its main mechanisms for deteriorating human health are not well-studied. Therefore, the work by the authors adds important insights in this direction. They demonstrate that chronic alcohol consumption increases cancer stem cell (CSC) populations and their stemness features including self-renewal capacity and further provide data for the underlying mechanism of action. The main findings of the work link chronic ethanol exposure to cancer stemness and aerobic glycolysis via the activation of the NFAT signalling pathway.

The work is of importance and interest. To deserve its place in the journal though, the the authors need to address some Major and Minor concerns  They are provided below in a detailed way:

Major

1. How was decided the non-cytotoxic dose of EtOH (100 mM)?

2. Why did you choose exactly these cell lines: SCC9 and UM6? How about normal cell lines for comparison of the obtained results and providing selectivity of alcohol action?

3. Figure 1: what kind of cell counting you have performed and how are you sure that these results are not biased? Why are standard tests like MTT and others not applied for cell viability and cytotoxicity investigations?

4. Cell colonies in Fig.1B are not visible well?

5. Where is the statistical data presented on the graphs? No one asterisk is visible, just is mentioned in the Figure caption.

6. Figure 2B: The Aldefluor flow cytometry-based assay dot plots are not visible at all. Please, provide quantitation of the FACS data too in a Table or by a graph.

7. Figure 2C: The RT-qPCR data do not have statistical analysis? Could you provide and explain why you prefer relative mRNA expression rather than fold change?

8. The representative images of tumour spheres formed by the OSCC cells shown on the right of the bar graph in Figures 2D and F are not visible.

9. Figure 2G does not have statistical data analysis?

10. Figure 5B and C lack statistical data. The representative images are not of good quality. Please, provide higher resolution and better quality images. As well as bars are missing to allow the presentation of the size.

11. Figure 6 misses statistical data.

Minor

1. A correlation between the applied concentrations of EtOH and daily human consumption would be very important for the discussion part of the work as well as for proving its relevance and significance for the field. Please, provide this correlation for the applied: 25-300 mM for 3 and 7 days of EtOH on the cells and link it to human daily consumption rates.

2. The authors need to proofread the paper for typos, spelling errors and grammar issues.

Reviewer 3 Report

This is a revised manuscript of an original manuscript in which the authors described the outcomes and discussion of their study, which focused on investigating the effects of chronic alcohol exposure on glycolytic processes and glycolytic enzymes including NFAT in  two oral/ l/oropharyngeal squamous cell carcinoma  (OSCC) cell lines. In their original manuscript they provided strong evidence that chronic alcohol exposure in these the two OSCC cell lines they tested, led to promotion of cancer stemness and glycolysis by activating NFAT signaling mechanisms. In their original submission they provided strong evidence in support of their conclusion. From glycolytic stand point, the outcomes of their studies include very interesting findings regarding how chronic alcohol exposure could regulate key glycolytic steps as well as regulate cancer stemness. The manuscript was relatively well prepared with good data presentation.  However, all the reviewers of their original manuscript identified many significant weaknesses and concerns, which together caused the three reviewers to recommend major revision.

In this revised manuscript, the authors did excellent job by positively responding to the critiques of all the three reviewers. The authors have addressing most of the key and relevant weaknesses and concerns. For example. they have consistently explained the purpose of this study and offered better explanation regarding why they performed their study in two OSCC cell lines. In most instances, they mostly agreed with the reviewers in most areas and modified their manuscript to reflect the reviewers’ recommendations and/or suggestions. Furthermore, where appropriate, the authors have cited published papers including their own and others, to support their explanation. Thus, the authors have been positively responsive to the reviewers.

The revised manuscript emphasizes the relevant of this study and its biochemical and physiological important in vitro, which will most likely, set the stage for a near future investigation of chronic exposure in animals. Their experimental designs, data presentation, statistical analysis, and data interpretation conform to the standards of this journal. Their conclusion are supported by their findings. This revised manuscript has highly significant quality as compared to their original submission.

Chronic alcohol exposure in mammals is known to lead to major impact on key metabolic processes including glycolysis and other energy generation systems. However, our knowledge on the effects of chronic alcohol exposure in OSCC and the mechanisms involved were not well understood. The findings of this work that are reported in this revised manuscript, clearly will contribute to our knowledge on chronic alcohol exposure in cancer cells. Furthermore, the outcomes and conclusion of their study  implicate activation of the NFAT signaling pathway as a key regulatory process by which chronic exposure of high concentration of concentration could impact on cancer cell stemness.  In my opinion, the revised manuscript has significantly been improved and will most likely contribute important information to the field. 

Author Response

We greatly appreciate the reviewer’s favorable comments on our revision.

Reviewer 4 Report

The authors did a nice job showing the roles of cancer stemness and metabolic reprogramming 32 in the molecular mechanism of alcohol-mediated carcinogenesis. However, there are some concerns regarding the experiment design and the presentation.

1) First of all I don't see how the experiment design mimics the chronic exposure of alcohol to OSCC? I mean the concentration and the time period are too high. Maybe if the title changed from " Oral/Oropharyngeal Squamous Cell Carcinoma" to " Oral/Oropharyngeal Squamous Cell Carcinoma cell lines" the model will be more suitable.

2) The cell number used in the tumor sphere formation assay is a bit confusing in the methods section. The same applies to the time period, 6-10 days is not specific enough making it difficult for reproducibility. 

3) In the migration assay in the methods section, there is no need to cite 4 papers of yours to explain the technique. Just one with a detailed description will be enough to avoid self-citation.

4) How do you explain the tumor regression in the in-vivo model even with EtOH-treated cells with increased tumorigenic ability?

Round 2

Reviewer 1 Report

The authors had responded to all questions. However, the knockdown efficacy of NFATc2 identified by western blot and the clinical relevant of NFATc2 in oral cancer patients with chronic alcohol consumption are critical data in present study. Without these evidences, the study is unconvincing. I disagree that this manuscript meets the criterion for acceptance.

Reviewer 2 Report

The authors have addressed the comments from Review round one.
Please, double-check and proofread your text again.

Reviewer 4 Report

Thank you no further comments

Round 3

Reviewer 1 Report

The evidences and interpretations provided by the authors do not convince me, so I don't approve the publication of this manuscript.

Author Response

We regret that the reviewer was not convinced by our study.